# Modelling and analysis of cAMP-induced mixed-mode oscillations in cortical neurons: Critical roles of HCN and M-type potassium channels

**Matteo Martin, Morten Gram Pedersen** *

Department of Information Engineering, University of Padova, Padova, Italy

* mortengram.pedersen@unipd.it

**Data Availability Statement:** All code written in support of this publication is available at https://researchdata.cab.unipd.it/1194.

## Abstract

Cyclic AMP controls neuronal ion channel activity. For example hyperpolarization-activated cyclic nucleotide–gated (HCN) and M-type $K^+$ channels are activated by cAMP. These effects have been suggested to be involved in astrocyte control of neuronal activity, for example, by controlling the action potential firing frequency. In cortical neurons, cAMP can induce mixed-mode oscillations (MMOs) consisting of small-amplitude, subthreshold oscillations separating complete action potentials, which lowers the firing frequency greatly. We extend a model of neuronal activity by including HCN and M channels, and show that it can reproduce a series of experimental results under various conditions involving and inferring with cAMP-induced activation of HCN and M channels. In particular, we find that the model can exhibit MMOs as found experimentally, and argue that both HCN and M channels are crucial for reproducing these patterns. To understand how M and HCN channels contribute to produce MMOs, we exploit the fact that the model is a three-time scale dynamical system with one fast, two slow, and two super-slow variables. We show that the MMO mechanism does not rely on the super-slow dynamics of HCN and M channel gating variables, since the model is able to produce MMOs even when HCN and M channel activity is kept constant. In other words, the cAMP-induced increase in the average activity of HCN and M channels allows MMOs to be produced by the slow-fast subsystem alone. We show that the slow-fast subsystem MMOs are due to a folded node singularity, a geometrical structure well known to be involved in the generation of MMOs in slow-fast systems. Besides raising new mathematical questions for multiple-timescale systems, our work is a starting point for future research on how cAMP signalling, for example resulting from interactions between neurons and glial cells, affects neuronal activity via HCN and M channels.

## Author summary

Neurons use the frequency of electrical signals called action potentials to encode information, and various messenger systems interact with ion channels to control this so-called

**Funding:** The author(s) received no specific funding for this work.

**Competing interests:** The authors have declared that no competing interests exist.

firing frequency. Recent experimental recordings show that the intracellular messenger cAMP can induce mixed-mode oscillations (MMOs) consisting of small-amplitude, sub-threshold oscillations separating action potentials, which lowers the firing frequency greatly. We extend a recent mathematical model of neuronal electrical activity to investigate how MMOs occur from interactions between ion channels regulated by cAMP. Our simulations reproduce a range of experimental results, including cAMP-induced MMOs. We explain the model dynamics using modern geometrical methods that exploit the different timescales in the model. Our analyses show that the very slow dynamics of cAMP-regulated HCN and M ion channels is not crucial for creating MMOs, but rather that the cAMP-induced increase in their average activity is important. Our analyses suggest that both HCN and M channels are crucial for MMOs and controlling the firing frequency, which has implications for our understanding of how astrocytes control neuronal information processing. Moreover, our study raises new mathematical questions related to how super-slow dynamical variables modify MMOs.

## Introduction

Cyclic AMP (cAMP) is an ubiquitous second messenger involved in a wide range of intracellular signaling processes. In neurons, cAMP has been suggested to control excitability and electrical activity by regulating ion channel activity [1, 2]. Of particular interest for the present work, in cortical neurons cAMP activates hyperpolarization-activated cyclic nucleotide–gated (HCN) channels [3, 4], which mediate a depolarizing current, and M-type potassium (also known as KCNQ or Kv7) channels, which carry a hyperpolarizing current [5, 6].

Recent experimental findings from layer V pyramidal cells [4] demonstrated how $Ca^{2+}$-evoked ATP release from astrocytes modulates the action potential (AP) conduction speed and the neuronal membrane excitability through cAMP increase and HCN channels. It was shown that HCN activation changes spiking electrical activity consisting of regular AP firing, into complex electrical patterns of mixed-mode oscillations (MMOs), where subthreshold, small amplitude oscillations (SAOs) intersperse large amplitude oscillations (LAOs), i.e., complete APs. This shift leads to a large reduction in the inter-spike frequency due to the presence of SAOs. Similarly, cAMP-induced activation of M-type channels has been shown to lower the firing frequency in pyramidal cells by producing MMOs [5]. How cAMP through both hyper- and depolarizing channels can cause MMOs is far from trivial.

This work proposes and investigates the hypothesis that the experimentally observed phenomena in [4, 5] result from both HCN and M channels. To study this idea, the neuronal electrical activity is modelled by the introduction of HCN and M currents in the model presented in [4], which is an adaptation of a previous model [7] to pyramidal cells. This extended and optimized model replicates the experimental results presented in [4, 5]; in particular, it produces MMOs upon an increase in cAMP levels, in contrast to the simulations shown by Lezmy et al. [4]. To understand how M and HCN channels contribute to produce MMOs, we exploit that the model presents variables with different velocities, i.e., it is multiple-time scale dynamical system with one fast, two slow, and two super-slow variables. In this class of models, geometrical and mathematical approaches are used to explain temporal dynamics, and the mechanisms generating MMOs are increasingly well understood [8]. Using these mathematical tools, it has been possible to understand and explain MMOs observed in many cellular quantities, e.g., complex patterns of electrical activity in neurons [9–11], pituitary cells

[12–14], human beta cells [15], and cardiomyocytes [16, 17], as well as mixed-mode calcium oscillations [18], complex dynamics appearing from cell-to-cell interaction [19–21], etc.

We find that both HCN and M channels are important for creating MMOs in the model. However, this is not because of the dynamics of their gating variables, which operate on the super-slow timescale, since the model is able to produce MMOs even when HCN and M channel activity is kept constant. In other words, cAMP increases the average HCN and M channel activity to set the slow-fast subsystem in a region of parameter space where MMOs are produced by the slow-fast subsystem alone. We show that the slow-fast subsystem MMOs are due to a folded node, and we construct numerically the relevant geometrical objects that explain the detailed dynamics of the simulated MMOs.

## Results

### The model reproduces MMOs in various experimental conditions

We included HCN and M currents in a previous model of neuronal activity [4, 7] in order to simulate and investigate electrical activity under different experimental conditions [4, 5], which examined how the cAMP-dependent pathway depicted in Fig 1 influence neuronal behavior. This model was adapted to pyramidal cells by Lezmy et al. [4], who observed MMOs experimentally in this type of cortical neurons. Our simulations reproduce satisfactorily a range of biological results, and in particular exhibit MMOs as found experimentally. The Hodgkin-Huxley-type model is formulated as a set of ordinary differential equations and has five variables: the membrane potential ($V$) and the gating variables describing, respectively, inactivation of the fast $Na^+$ current ($h$) and activation of the slow $K^+$ ($s$), HCN ($r$) and M ($w$) currents. More details can be found in Materials and methods. Model parameters are given in Table 1.

### Elevated cAMP following A2aR activation induces MMOs

In the first experiment, the control condition is compared to the experiment where a stimulus is provided via CGS21680 (CGS) [4]. This drug activates A2aR and the downstream pathway, increasing the M and HCN currents, which can cause MMOs [4, 5].

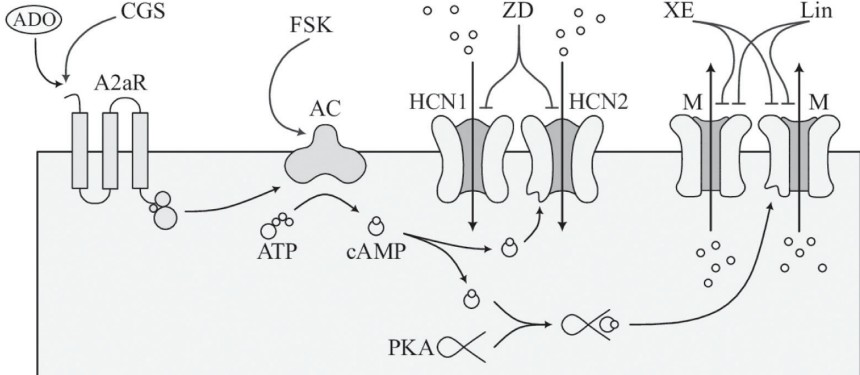

**Fig 1. Schematic representation of the considered pathways.** The scheme depicts how Adenosine 2A Receptor (A2aR) activation by extracellular Adenosine (ADO) increases cAMP levels via activation of adenylate cyclases (AC), which in turn increases the opening probability of HCN and M channels via direct binding and PKA activation, respectively. The experiments in [4, 5] target this pathway through specific drugs to understand how both M and HCN channels participate in the modulation of neuronal electrical activity. The chemical compounds employed are the A2aR agonist CGS21680 (CGS), the HCN channel inhibitor ZD7288 (ZD), the AC activator Forskolin (FSK), and the M channel antagonists XE991 (XE) and Linopirdine (Lin).

The system of ODEs is simulated for different intensities of applied current (Fig 2). For the lower values of $I_{App}$, MMOs are seen both in control conditions and when cAMP is raised by CGS, whereas for $I_{App}$ equal to 250 and 300 $\mu$A/cm$^2$, MMOs are seen only in the presence of GCS, while regular AP firing is obtained in the control condition, as seen experimentally [4]. Thus, the CGS-induced increases of the M and the HCN currents expand the interval of $I_{App}$ values for which the system exhibits MMOs, passing from [81, 223] $\mu$A/cm$^2$ to [62, 310] $\mu$A/cm$^2$.

The activation of the cAMP pathway augments the hyperpolarizing effect of the M current during the AP (Fig 3, turquoise, lower panels). In contrast, activation of HCN channels becomes important after the AP hyperpolarization phase (Fig 3, turquoise, lower panels). However, as will be shown in the following, although M and HCN currents are important for positioning the system so that MMOs can appear, it is not their dynamics that cause the SAOs. Rather, when the depolarizing HCN currents destabilize the resting membrane potential (Fig 3, upper panels), the slow K$^+$ current $I_{KS}$ counteracts the depolarization (Fig 3, blue, middle row), which in turn releases inactivation of the fast Na$^+$ current $I_{NaF}$ (Fig 3, orange, middle row). The interplay between $I_{KS}$ and $I_{NaF}$ causes subthreshold oscillations, until $V$ crosses the AP firing threshold.

## When HCN currents are blocked, cAMP can silence active cells

Lezmy et al. [4] tested how HCN currents influence neuronal responses by blocking HCN channels pharmacologically with ZD7288 (ZD) under control conditions, or when the cAMP pathway was activated by CGS. In the model, this corresponds to setting the HCN conductance ($g_{HCN}$) to zero and increasing the M channel conductance only. Fig 4, left panels, presents simulated electrical activity under ZD and ZD+CGS applications. For $I_{App}$ = 115 $\mu$A/cm$^2$, MMOs are observed in the absence of CGS and presence of ZD, as in the control condition (Fig 2), whereas the addition of both ZD and CGS turns the neuron silent because of the activation of the hyperpolarizing M current. For $I_{App}$ = 300 $\mu$A/cm$^2$ and in the presence of ZD, the increased hyperpolarizing effect of the M currents due to CGS changes continuous spiking into MMOs. Compared to the control case, the suppression of the HCN currents shifts the current range for which MMOs are observed from [81, 223] $\mu$A/cm$^2$ to [103, 233] $\mu$A/cm$^2$. Additional administration of CGS in the presence of ZD further right-shifts this interval to [118, 337] $\mu$A/cm$^2$. These simulations correspond to the

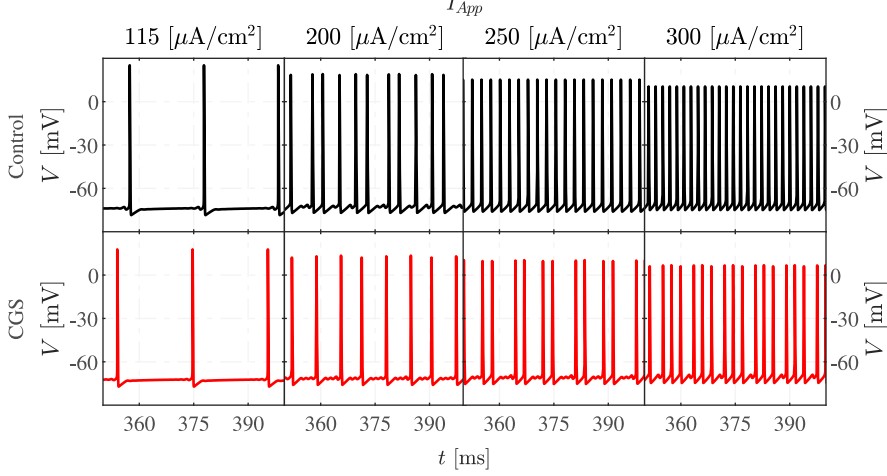

**Fig 2. Elevated cAMP promotes MMOs.** The figure presents simulated voltage traces under control conditions (first row, black curves) and when cAMP is elevated as in the experiments with CGS application (second row, red curves) at four different values of $I_{App}$.

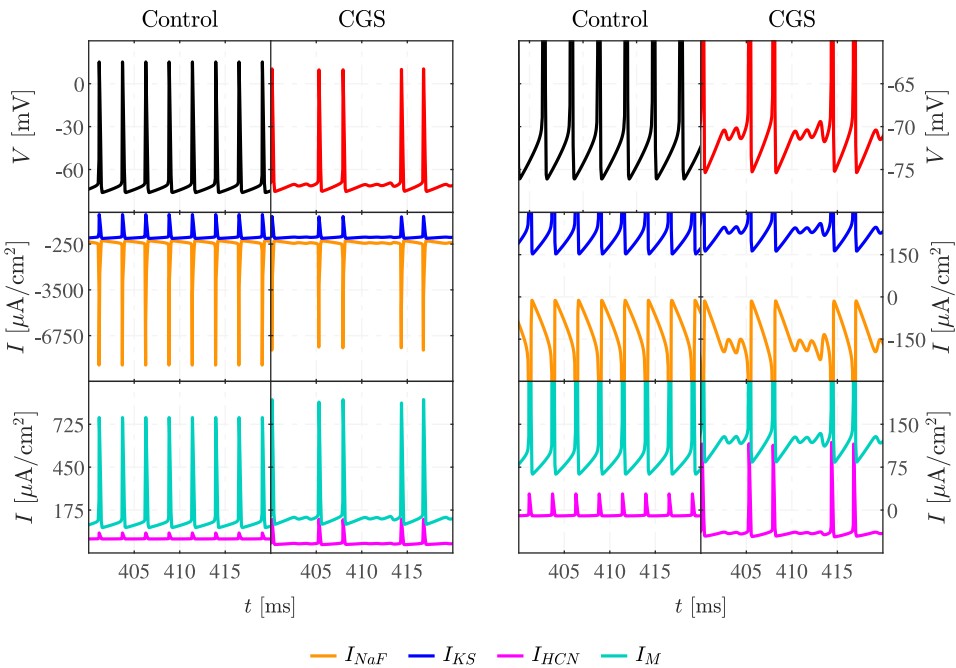

**Fig 3. Ionic currents during APs and MMOs.** Voltage (upper row; black and red curves) and principal currents (middle and bottom rows; see legend for interpretation of colors) in simulations with $I_{App}$ = 250 $\mu$A/cm$^2$, in control and CGS-stimulated conditions, are shown entirely in the left panels, whereas the right panels show zooms on the dynamics near the onset of APs and the SAOs involved in MMOs.

observation that in the presence of ZD and at low stimulus strength, A2aR-mediated signalling from astrocytes reduced the frequency and even stopped neuronal AP firing [4].

## Blocking M channels changes MMOs to spiking and increases the firing frequency

Arnsten et al. [5] blocked M channels with XE991 (XE) to investigate the roles of both M and HCN channels. We simulated this experiment by setting the M channel conductance to zero in the absence or presence of elevated cAMP (Fig 4, right panels). For $I_{App}$ = 115 $\mu$A/cm$^2$, the block of the M currents turned MMOs (Fig 2, control case) to continuous spiking (Fig 4). For $I_{App}$ equal to 300 $\mu$A/cm$^2$, XE lowered the AP amplitude and increased the firing frequency compared to the control scenario (Fig 2), and in the presence of CGS M-channel block changed MMOs to low-amplitude AP firing. M-channel inhibition allows the HCN channel to depolarize the membrane potential quickly after the AP hyperpolarization phase. The fast Na$^+$ channels do therefore not reactivate completely, which limits the following AP upstroke. The MMO oscillation regime, compared to the control case, narrows down to [65, 103] $\mu$A/cm$^2$ in the presence of XE. Combined XE and CGS application shifts this interval to [19, 80] $\mu$A/cm$^2$. Overall, the results in Fig 4 correspond to the experimental findings obtained in [5].

## Partial inhibition of M channels in stimulated conditions can change the signature of MMOs

To understand how M channels influence the observed dynamics with elevated cAMP, Forskolin (FSK), which raises the cAMP concentration, and the M-channel inhibitor Linopirdine were applied to the neurons [5].

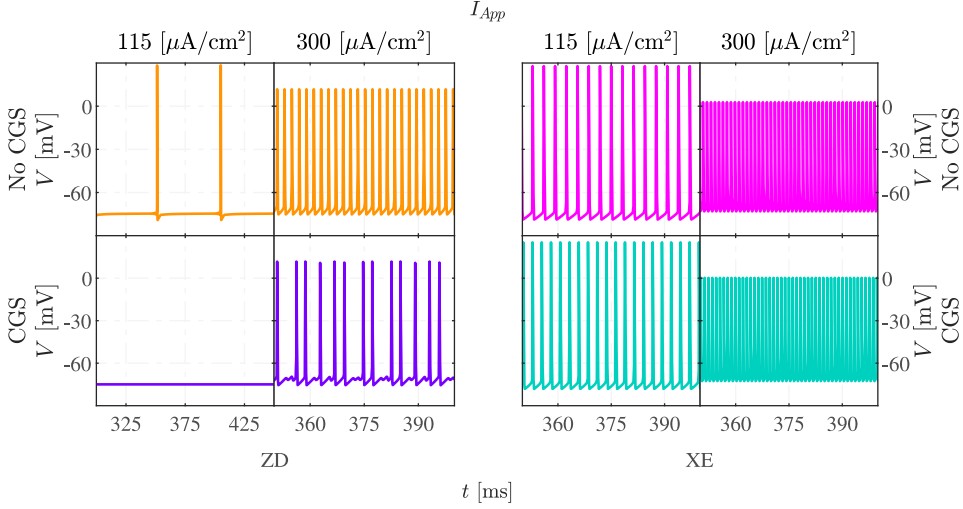

**Fig 4. Blocking HCN and M channels affects electrical patterns at basal and elevated cAMP levels.** The figure presents the model simulations using $I_{App}$ equal to 115 and 300 $\mu A/cm^2$, in the presence of either the HCN blocker ZD7288 (ZD; left panels) or the M current blocker XE991 (XE; right panels). The first row shows simulations without activation of the cAMP-dependent pathway, whereas those presented in the second row are for CGS application.

Fig 5 presents the model results after the administration of FSK alone, or in the presence of both FSK and Linopirdine. Forskolin increases both $g_M$ and $g_{HCN}$, whereas the combined administration of FSK and Linopirdine is modelled by a reduced increase in $g_M$ (see Table 2), i.e., the application of Linopirdine to an FSK-stimulated cell gives a net reduction of $g_M$. The partial inactivation of the M channel destabilizes the resting membrane potential and increases the excitability of the neuron. This effect explains the change in the MMO signatures in the presence of Linopirdine, with more LAOs and shorter sequences of SAOs compared to the scenario with fully active M channels.

## Firing frequency analyses

Fig 6 shows firing frequency curves under different experimental conditions for a range of $I_{App}$ values. The FF curve associated with the unstimulated neuron (black) splits into several parts.

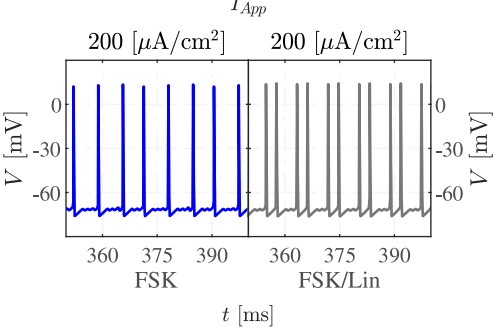

**Fig 5. M-channel inhibition changes the signature of MMOs in stimulated conditions.** Simulations of the model with raised cAMP in the absence (left, blue) or presence of partial M channel inhibition, as in the experiments with Forskolin (FSK) application in the absence or presence of Linopirdine (Lin) [5], are presented. Note how the grey trace presents more full APs (LAOs) and fewer subthreshold oscillations (SAOs).

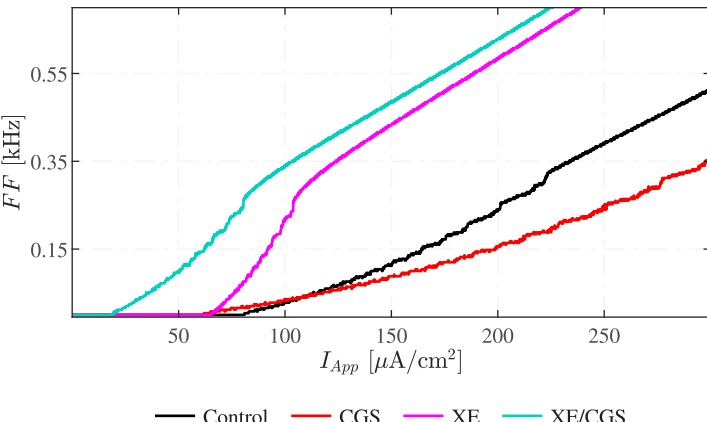

**Fig 6. Firing Frequency analyses.** The FF of the 5D model under various pharmacological stimulations at different levels of applied current. The black curve shows FFs for the unstimulated (control) condition. Red, magenta and turquoise curves are associated with GCS21680 (GCS; increased cAMP), XE991 (XE; M channel block), and simultaneous XE991+CGS21680 (XE/CGS) application, respectively.

For $I_{App}$ in the interval $[81, 223]$ $\mu$A/cm$^2$, the neuron exhibits MMOs. The neuron is silent if $I_{App}$ is below $81$ $\mu$A/cm$^2$, whereas for $I_{App} \in [223, 720]$ $\mu$A/cm$^2$, only LAOs persist, i.e., the cell fires regularly. The application of CGS activates A2aR, which raises cAMP levels, increasing both HCN and M currents. This modification changes the FF curve (red) by shifting its onset leftward and making it less steep. In fact, the unstimulated FF curve crosses the one obtained for CGS stimulation at $I_{App} \approx 113$ $\mu$A/cm$^2$. Above this value, the unstimulated case presents higher FF, while the stimulated one is more active for $I_{App}$ below the threshold. This crossing of the FF curves agrees with the results by Lezmy et al. [4].

Fig 6 shows the FF results for two additional conditions. M currents are inhibited in both these cases, mimicking the application of XE in the absence (magenta) or presence (turquoise) of the A2aR agonist CGS. Considering the XE-treated neuron, the associated FF curve is steeper compared to the control and the CGS-treated cases. In fact, under M-channel inhibition, the neuron generates oscillatory phenomena for lower $I_{App}$, due to the lack of the stabilizing, hyperpolarizing M current. For $I_{App}$ in $[65, 81]$ $\mu$A/cm$^2$, the neuron is silent under control conditions, while it undergoes MMOs with a FF approximately proportional to $I_{App}$ if treated with XE. If the neuron pre-treated with XE undergoes CGS stimulation, the FF curve shifts further to the left. The increase in the depolarizing HCN channel conductance explains this movement, since the neuron is more excitable. The comparison between the curves presented in Fig 6 and the experimental FF curves [4, 5], strongly suggests that both M and HCN currents are required to explain how cAMP controls the FF in cortical neurons.

## Dynamical system analyses of the slow-fast subsystem

To understand how MMOs appear in the 5D model simulated above, we analyse the three-dimensional slow-fast ($V$, $h$, $s$) subsystem of the 5D model, which allows us to apply standard methods regarding folded singularities [8]. Exploiting the fact that the M and HCN channels have slower dynamics than the other variables in the model, the activation variables of M ($w$) and HCN ($r$) channels are considered as parameters in the slow-fast ($V$, $h$, $s$) subsystem (the "3D model" in the following). The reduction steps are illustrated in the section Model reduction in Materials and methods. Fig 7 presents the 3D model simulation of the experiment with CGS stimulation, corresponding to the 5D model simulation shown in Fig 2. Crucially, the

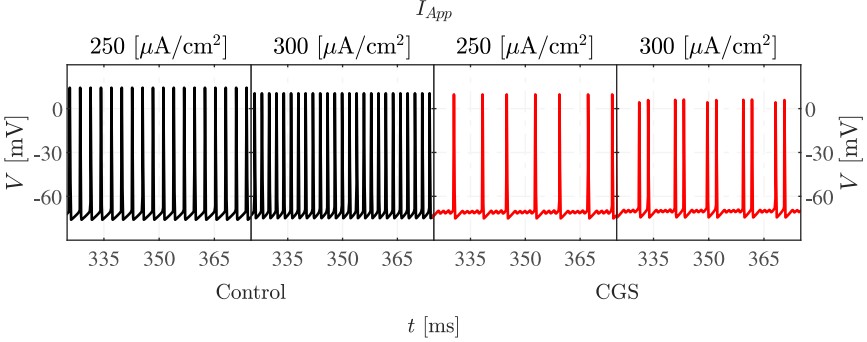

**Fig 7. 3D model voltage traces.** Simulations of the 3D model under control (left, black) and CGS-stimulated (right, red) conditions, compare with Fig 2.

dynamics of HCN and M channels are not strictly necessary in order to observe MMOs, which is a result of the dynamics of the 3D slow-fast subsystem. The most significant difference between the 5D (Fig 2) and the 3D (Fig 7) simulations is seen in the signatures of the MMOs presented by the two systems. Specifically, for the same value of $I_{App}$, the 3D model shows fewer LAOs and more SAOs than the 5D model. For example, for $I_{App} = 250 \,\mu$A/cm$^2$, the full system presents 2 LAOs followed by 2 or 3 SAOs, whereas the 3D model experiences 1 LAO followed by 3 or 4 SAOs. This variation is caused by the simplification of the model.

In order to be able to use the same slow-fast subsystem bifurcation diagrams in the presence and absence of cAMP-mediated activation of M and/or HCN channels, we introduce the modified slow variables (where $cAMP = 0$, respectively $cAMP = 1$, indicate absence, respectively presence, of cAMP-mediated channel activation)

$$\tilde{w} = w\left(1 + cAMP\frac{\Delta g_M}{g_M}\right), \qquad \tilde{r} = r\left(1 + cAMP\frac{\Delta g_{HCN}}{g_{HCN}}\right), \qquad (1)$$

which are then used as bifurcation parameters in the slow-fast subsystem. In these expressions, $\Delta g_M$ and $\Delta g_{HCN}$ are the increases in, respectively, M channel and HCN conductances caused by increased cAMP. Using this formulation, the value of $cAMP$ is completely incorporated into the parameters $\tilde{w}$ and $\tilde{r}$, see Materials and methods for further detail.

## Slow-fast subsystem bifurcation diagrams

When projecting the trajectory of the 5D model onto the $(\tilde{w}, \tilde{r})$ plane, the model trajectory evolves close to a straight line $\tilde{r} = m\tilde{w} + q$ (Fig 8B). We can therefore construct one-parameter bifurcation diagrams (1P-BD) for the 3D slow-fast subsystem with bifurcation parameter $\tilde{w}$, and $\tilde{r} = m\tilde{w} + q$ constrained to lie on the identified straight line. However, the line depends on the experimental condition that is simulated, and moves upwards when cAMP is elevated (Fig 8B): Both $\tilde{w}$ and $\tilde{r}$ increase since cAMP introduces a shift in these variables by construction.

Fig 8A shows the computed 1P-BD for the slow-fast $(V, h, s)$ subsystem. The results show the existence of a unique equilibrium point. For high M-channel activation $\tilde{w}$, the equilibrium is stable. As $\tilde{w}$ is reduced (and $\tilde{r}$ is increased), the equilibrium loses stability in a Hopf bifurcation (HB). From the HB, unstable periodic solutions emerge. On the other hand, at very low $\tilde{w}$, the branch of unstable equilibria is surrounded by a branch of stable limit cycles, and as $\tilde{w}$ is increased, these lose stability as they go through a cascade of period-doubling bifurcations

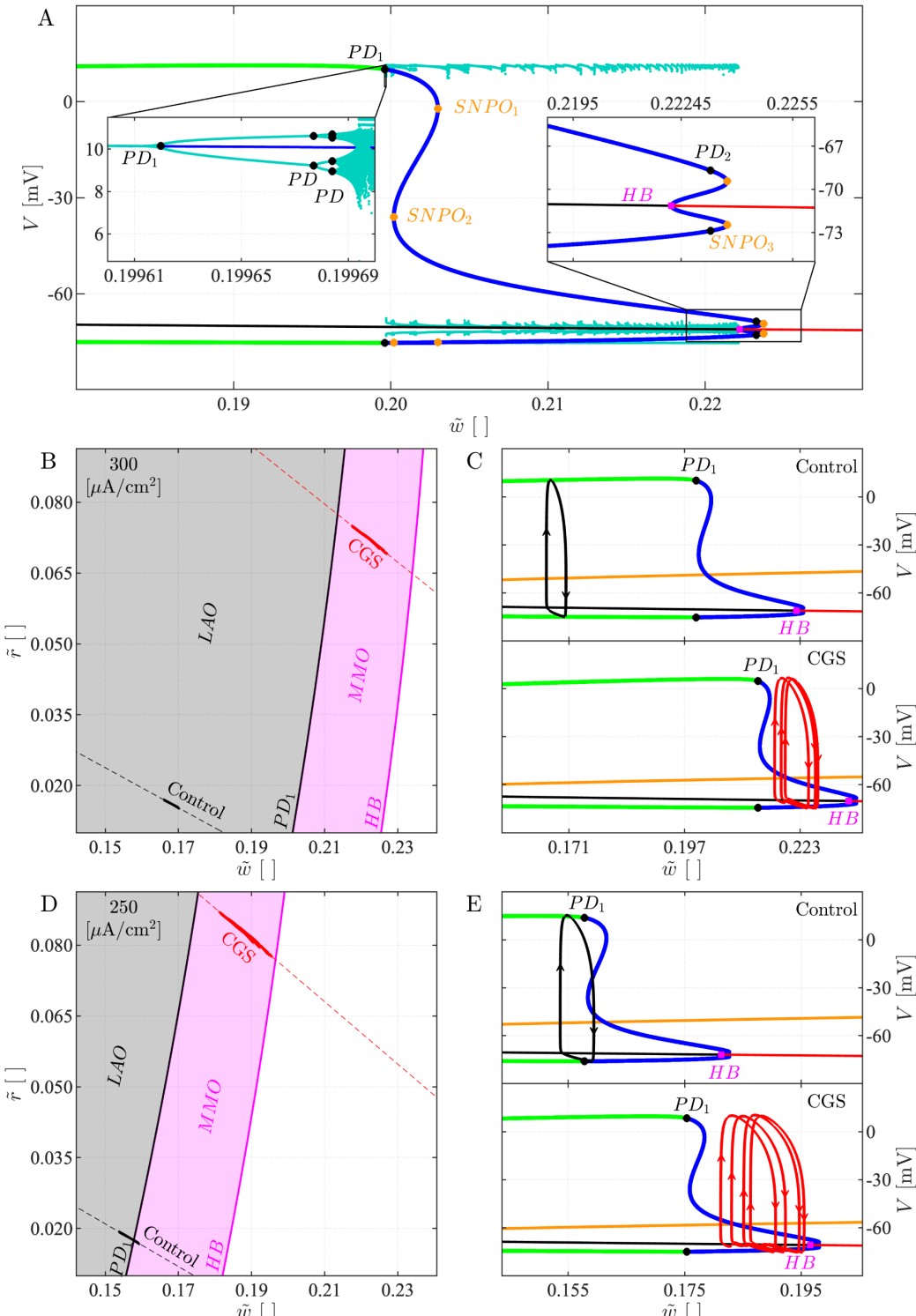

**Fig 8. Slow-fast 3D subsystem bifurcation diagrams.** *A*: 1P-BD of the slow-fast subsystem for $I_{App} = 300 \, \mu A/cm^2$ in the control condition with $\tilde{w}$ as parameter and $\tilde{r}$ constrained to the dashed, black line in panel B. The red (respectively, black) curve indicates stable (unstable) equilibria, and the green (blue) curves are minima and maxima of stable (unstable) periodic solutions. The inset on the right shows a zoom on the region where the equilibrium changes stability. The inset on the left provides a zoom around the period-doubling cascade. HB: Hopf bifurcation, SNPO: saddle-node bifurcation of periodic orbits, PD: period doubling bifurcation. *B*: $(\tilde{w}, \tilde{r})$ plane with projections of 5D model simulations for $I_{App} = 300$

 

$\mu$A/cm$^2$ in control (full, black curve) and CGS-stimulated (red) conditions. The dashed lines indicate the linear approximations $\tilde{r} = m\tilde{w} + q$. The background shows the 2-parameter bifurcation diagram (2P-BD) for the slow-fast subsystem with $(\tilde{w}, \tilde{r})$ as parameters, obtained by following the most relevant bifurcations found in the 1P-BD (panel A). C: BD as in panel A with 5D simulation projected onto the $(\tilde{w}, V)$ plane for the control (upper) and CGS-stimulated (lower) scenarios. The orange curve is the $\tilde{w}$ nullcline. D and E: As panels B and C, but for $I_{App} = 250\,\mu$A/cm$^2$.

(PDs). Following saddle-node of periodic orbit (SNPO) bifurcations, the branch of (unstable) limit cycles eventually connects to the HB point. Between PD$_1$ and HB, the system exhibits MMOs. These do not appear as a direct result of the PDs, but rather when the chaotic attractor makes a sudden discontinuous jump slightly to the right of PD$_1$ at $\tilde{w} \approx 0.19969$.

Reintroducing the super-slow dynamics of $\tilde{w}$ (and $\tilde{r}$), we see that for $I_{App} = 300\,\mu$A/cm$^2$ in the control condition, the system follows a stable periodic orbit without exhibiting SAOs (Fig 8C, upper). In the case with CGS stimulation, the simulated trajectory is fully contained within the (3D slow-fast subsystem) MMO region (Fig 8C, lower): $\tilde{w}$ decreases during the SAOs, but increases during the LAOs (APs), so that $\tilde{w}$ remains in the interval with MMOs. Geometrically, the explanation is that in the presence of cAMP ($cAMP = 1$), the $\tilde{w}$ nullcline is moved down and to the right (Fig 8C), so that the system stalls further to the right in the MMO region, rather than in the LAO region as with $cAMP = 0$.

The behavior with lower applied current ($I_{App} = 250\,\mu$A/cm$^2$) is interesting. Here, the slow-fast subsystem bifurcations occur at lower values of $\tilde{w}$. Due to the evolution of the dynamical variables $\tilde{w}$ and $\tilde{r}$, the 5D system enters the region with slow-fast subsystem MMOs (Fig 8D). However, the system does not stay long enough in this region for SAOs to appear, since $\tilde{w}$ decreases for low $V$ and the trajectory moves to the left of PD$_1$ where an AP appears (Fig 8E, upper). With CGS stimulation ($cAMP = 1$; Fig 8E, lower), as in the case with $I_{App} = 300\,\mu$A/cm$^2$, the trajectory lies completely in the MMO region since the $\tilde{w}$ nullcline lies lower and to the right, compared to the control case ($cAMP = 0$).

Relaxing the constraint $\tilde{r} = m\tilde{w} + q$ permits construction of the two-parameter $(\tilde{w}, \tilde{r})$ BD (2P-BD) of the slow-fast $(V, h, s)$ subsystem by following the main bifurcations shown in the 1P-BD (Fig 8B and 8D). MMOs are observed only if the entire 5D model trajectory belongs to the MMO region. The control condition with $I_{App} = 250\,\mu$A/cm$^2$ is a borderline scenario. In fact, as discussed above, the trajectory crosses the PD$_1$ curve but presents no SAOs (Fig 2). Altogether, the retrieved 2P-BD for the 3D slow-fast subsystem reflects the full 5D model behavior well, and suggests that the average activation level of HCN and M channels is sufficient to predict the type of activity. For example, activation only of M channels corresponds to a right-shift in the $(\tilde{w}, \tilde{r})$ plane, which could move the system from the MMO region to the silent region to the right of the HB curve, as seen in Fig 4, left panels. Vice versa, blocking M channels corresponds to a left-shift, which can cause the system to go from the MMO to the LAO region, as seen in Fig 4, right panels, compared to Fig 2.

Similar reasoning can explain why, in the presence of the M current inhibitor XE, the addition of CGS (XE+CGS) shifts the FF curve towards the left (Fig 6). Mathematically, XE application corresponds to projecting the full-system trajectory onto the $\tilde{r}$ axis in the $(\tilde{w}, \tilde{r})$ plane. CGS addition is modelled by setting $cAMP = 1$, which increases $\tilde{r}$ (corresponding to activation of HCN channels) compared to the CGS-free case. For low values of $I_{App}$ (e.g., $I_{App} = 85\,\mu$A/cm$^2$), the 2P-BD looks similar to Fig 8B and 8D, but the different regions are located further to the left and shifted upwards so that the $\tilde{r}$-axis goes through the MMO region for a range of $\tilde{r}$ values spanning those observed in simulations modelling XE application. Hence, the system is exhibiting MMOs in this case. GCS application moves the system vertically along the $\tilde{r}$ axis into the LAO region where simple APs are observed. For even lower currents (e.g., $I_{App} = 50$

 

$\mu$A/cm$^2$), the different regions of the 2P-BD are shifted even further to the left and up so that the XE case is in the silent (white) region without CGS, but in the MMO region for when CGS is added in addition to XE, which again corresponds to moving along the $\tilde{r}$ axis.

The derived BDs can be interpreted biophysically as follows. As the depolarizing HCN channels activate (higher $\tilde{r}$), and the M channels become less active (lower $\tilde{w}$), the cell's excitability increases, facilitating AP generation. This observation explains why stable periodic orbits occur at a low $\tilde{w}$ and high $\tilde{r}$. Instead, when $\tilde{w}$ increases and $\tilde{r}$ decreases, the excitability of the model reduces. In addition, the more pronounced M channel activation is, the more SAOs appear, until the equilibrium corresponding to the resting potential eventually becomes stable for high $\tilde{w}$ values.

## M currents are necessary to reproduce the experiments

The model parameters are taken from [4, 7, 22]. However, the HCN reversal potential, $E_{HCN}$, the M-current conductance, $g_M$, and the increase in this conductance due to cAMP, $\Delta g_M$, are fine-tuned to satisfy both physiological and mathematical constraints to reproduce the electrical activity observed in the experimental study of [4] (Fig 2). The physiological restrictions require $E_{HCN}$ to be within the range [−50, −20] mV [23], while the cAMP-induced increase in M-current conductance is less than the conductance in control conditions, i.e., $\Delta g_M \leq g_M$ [5, 24]. The mathematical constraints should guarantee the reconstruction of the experimentally observed voltage traces [4]. That is, we look for a combination of ($E_{HCN}$, $g_M$, $\Delta g_M$) that generates spiking in control condition but MMOs when cAMP is raised, for example after CGS application [4].

By fixing $E_{HCN}$ and $g_M$, the modification of $\Delta g_M$ only affects the dynamics in the stimulated condition. As seen in the 2P-BD with bifurcation parameters $I_{App}$ and $\Delta g_M$ for $g_M$ = 50 mS/cm$^2$ and $E_{HCN}$ = −50 mV (Fig 9), higher values of $\Delta g_M$ expand and right-shift the interval where the model activates MMO at elevated cAMP. Similar results were found for other values of $g_M$ and $E_{HCN}$. In order for the model to produce, at some $I_{App}$, MMOs at elevated cAMP but spiking in control conditions, the area with MMOs at high cAMP must go further to the right than the MMO area for the control case. In Fig 9 this happens only for $\Delta g_M > 11$ mS/cm$^2$, and as $\Delta g_M$ increases the interval of $I_{App}$ values where spiking is seen in control condition but MMOs in stimulated condition widens. We used $\Delta g_M = g_M = 50$ mS/cm$^2$.

Summarizing, if cAMP does not affect the M current but HCN channels only, the model is not able to reproduce the experimental finding that CGS via cAMP elevation switches spiking electrical activity to MMOs.

## Geometry of mixed-mode oscillations

The 5D model is a three-time scale dynamical system as noted above. Its 3D slow-fast subsystem presents one fast ($V$) and two slow ($h$ and $s$) variables. We now explain the local and global dynamics of this 3D system with particular attention to the origin of MMOs. For a brief introduction to the underlying theory see Materials and methods, and, for in-depth expositions, refs. [8, 25].

The critical manifold $\mathcal{C}_0$ (Fig 10) is of great importance in order to understand the dynamics of the system. It is defined as the set of points ($V$, $h$, $s$) where $dV/dt = 0$, i.e., the set of equilibrium points for the fast subsystem of the 3D model. Points on $\mathcal{C}_0$ are said to be attracting or repelling if they are so when interpreted as fast-subsystem equilibrium points. For our model, the critical manifold can be expressed explicitly as a graph,

$$\mathcal{C}_0 = \{(V, h, s) \in \mathbb{R}^3 | s = \gamma(V, h)\}. \tag{2}$$

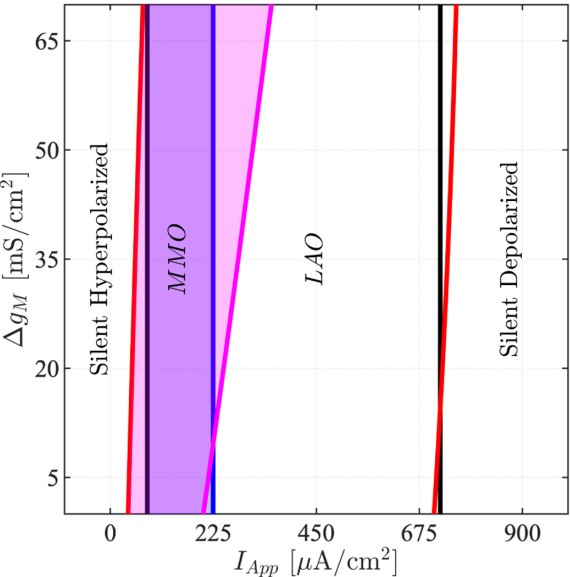

**Fig 9. Model dynamics depends on the degree of cAMP-activation of M channels.** Two-parameter BD with bifurcation parameters $(I_{App}, \Delta g_M)$ for $g_M = 50$ mS/cm$^2$, $E_{HCN} = -50$ mV, $g_{HCN} = 23$ mS/cm$^2$, and $\Delta g_{HCN} = 12$ mS/cm$^2$ (the default values used throughout the paper, see Table 1). Black and red curves indicate respectively HBs in the control conditions ($cAMP = 0$) and with raised cAMP ($cAMP = 1$). The curves in blue and magenta represent, similarly, the PDs (PD$_1$ in Fig 8) in the control and stimulated cases. The combination of $I_{App}$ and $\Delta g_M$ where MMOs occur is highlighted with shaded blue (control) and magenta (elevated cAMP) areas.

We find, in the physiologically relevant region, that $\mathcal{C}_0$ has a folded structure with two folds, $\mathcal{L}^{-/+}$. These folds split $\mathcal{C}_0$ into attracting and repelling sheets, denoted respectively $\mathcal{S}_a^{-/+}$ and $\mathcal{S}_r$. Starting from $V$ close to the neuron's resting potential, $\mathcal{C}_0$ is thus decomposed as $\mathcal{S}_a^- \cup \mathcal{L}^- \cup \mathcal{S}_r \cup \mathcal{L}^+ \cup \mathcal{S}_a^+$.

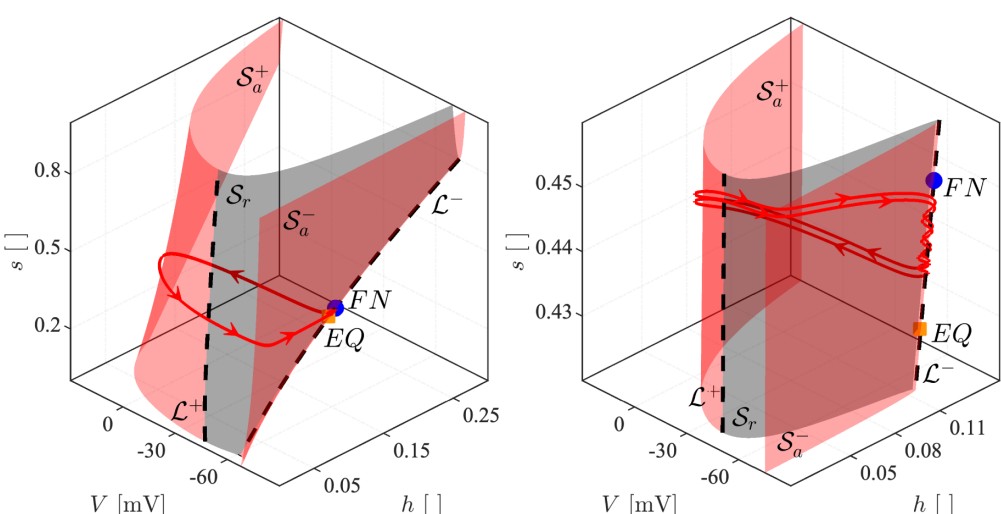

**Fig 10. Critical manifold.** This figure presents the 3D-model critical manifold $\mathcal{C}_0$ with $I_{App} = 250\ \mu$A/cm$^2$ in the CGS-stimulated case. The red curve is the corresponding trajectory from Fig 7. Attracting ($\mathcal{S}_a^{\pm}$) and repelling ($\mathcal{S}_r$) submanifolds are presented with red and grey shaded surfaces. The black dashed curves correspond to the fold lines $\mathcal{L}^{\pm}$. The folded node and the unstable fixed point are presented using a blue circle and an orange square, respectively. The right panel shows a zoom on the region near the trajectory.

At the fold, the system becomes singular (see Materials and methods and [8]). To understand the dynamics near the fold, the system is therefore desingularized, and it turns out that equilibrium points of the desingularized system on a fold, so-called folded singularities, play a crucial role in understanding the origin of SAOs [8]. We find that the model possesses a folded node (FN) singularity, which is well-known to cause SAOs [8].

Let $0 < \epsilon \ll 1$ denote the time scale separation between the fast $V$ variable and the slower $h$ and $s$ variables, see also Materials and methods. The following analyses assume that the model's initial conditions are located near $\mathcal{S}_a^-$.

Assuming stability of the 3D model equilibrium, i.e., for all $(\tilde{w}, \tilde{r})$ located on the right of the HB curve in Fig 8, the 3D slow-fast subsystem trajectory evolves dominated by the slow flow constrained to an $\mathcal{O}(\epsilon)$ perturbation of the attracting submanifold $\mathcal{S}_a^-$, denoted $\mathcal{S}_{a,\epsilon}^-$, and approaches the stable equilibrium in infinite time.

Instead, if the equilibrium point is unstable, the dynamics depends on how the system approaches the fold $\mathcal{L}^-$. For parameter combinations in the spiking region, e.g., the control condition shown in Figs 7 and 11 (upper panels), the system's trajectory evolves constrained to $\mathcal{S}_{a,\epsilon}^-$ until it reaches the fold curve $\mathcal{L}^-$ at a *regular jump point* [8, 25], where it then switches dynamics and moves to $\mathcal{S}_{a,\epsilon}^+$ via its fast flow. In Fig 11, this corresponds approximately to crossing $\mathcal{L}^-$ to the left of the FN. On $\mathcal{S}_{a,\epsilon}^+$ it again evolves according to the slow flow. When it reaches $\mathcal{L}^+$, the system jumps to $\mathcal{S}_{a,\epsilon}^-$, and then the above steps repeat, creating relaxation oscillations.

If, on the other hand, the system approaches $\mathcal{L}^-$ near the folded node, more precisely in the *funnel region* (the area with blue shading lines in Fig 11), SAOs are produced as the system passes from $\mathcal{S}_{a,\epsilon}^-$ to $\mathcal{S}_{r,\epsilon}$ [8, 15, 25]. The funnel is divided into *rotational sectors*, and the number of SAOs depends on the sector in which the folded node is approached. The trajectory eventually jumps to $\mathcal{S}_{a,\epsilon}^+$, corresponding to the onset of an AP, i.e., a LAO, and then follows the slow

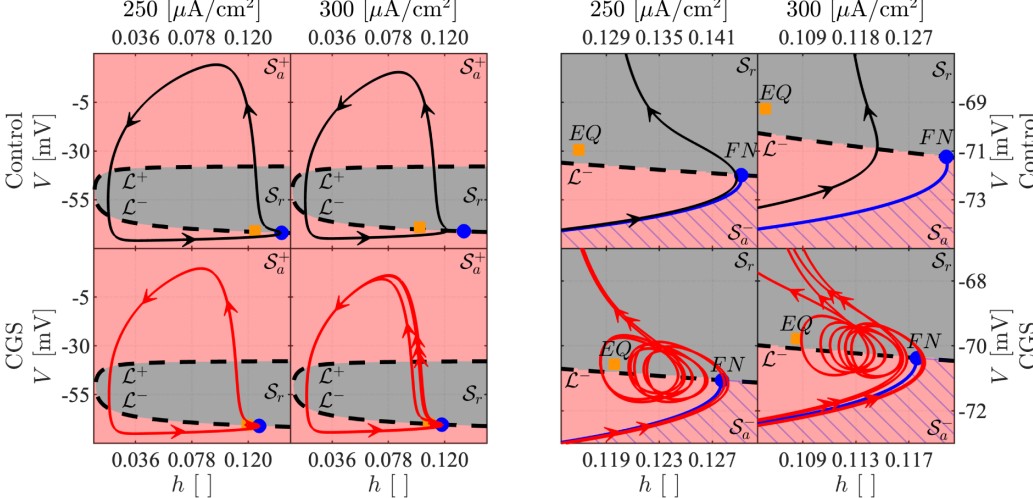

**Fig 11. Critical manifold and folded singularities of the 3D model.** The presented phase-plane plots show the trajectory of the 3D model with parameters as in Fig 7 projected onto the $(h, V)$ plane for the control case (black curve, upper panels) and in the presence of CGS (red, lower), observed globally (left) and locally with a zoom on the region where SAOs appear for the CGS-stimulated case (right). The dashed black curve represents the fold $\mathcal{L}^{-/+}$ of the critical manifold. The red and gray shaded regions correspond to the attracting, $\mathcal{S}_a^{-/+}$, and repelling, $\mathcal{S}_r$, submanifolds, respectively. The unstable equilibrium point is given by the orange square, while the blue dot indicates the FN. The blue curve is the *strong canard* that together with the fold line delimits the funnel region indicated by the blue horizontal shading lines. The arrows indicate the direction of the flow.

flow until it reaches $\mathcal{L}^+$ and jumps back to $\mathcal{S}^-_{a,\epsilon}$. Depending on the return mechanism, the trajectory may reenter the funnel region, producing single LAOs separated by SAOs, or it may reach $\mathcal{L}^-$ at a regular jump point, thus producing two or more LAOs separated by SAOs, as in Figs 7 and 11 with CGS for $I_{App}$ = 250 and $I_{App}$ = 300 $\mu$A/cm$^2$, respectively. In other words, the difference with respect to the number of LAOs between $I_{App}$ = 250 $\mu$A/cm$^2$ and $I_{App}$ = 300 $\mu$A/cm$^2$ is due to differences in the return mechanisms. In the former case, the orbit is sent back into the funnel after each LAO, whereas for $I_{App}$ = 300 $\mu$A/cm$^2$ the orbit does not enter the funnel after the first LAO but only after the second LAO.

In more detail, the dynamics shown in Figs 7 and 11 can be understood as follows. Each rotational sector $R_i$ is bounded by two *canards*, special solutions that connect the attracting and repelling sheets of the slow manifold, which we denote $\xi_{i-1}$ and $\xi_i$, where $i$ indicates the number of SAOs exhibited by the canard. The so-called *strong canard* $\xi_0$ delimits, together with the fold $\mathcal{L}^-$, the funnel region. For more detail on canards, see Multiple time scale dynamical system.

In Fig 7, the system at $I_{App}$ equal to 250 and 300 $\mu$A/cm$^2$ evolves with *signatures* $1^4 1^3$ and $2^4 2^3$, respectively. In the former case, the model generates 1 LAO followed by 4 or 3 SAOs. Thus, the return mechanism projects alternatingly the trajectory into $R_4$ delimited by $\xi_3$ and $\xi_4$, (Fig 12, lower left panel, black segment), and into $R_3$ delimited by $\xi_2$ and $\xi_3$, (blue segment). For $I_{App}$ = 300 $\mu$A/cm$^2$, 2 LAOs occurs, followed by 3 or 4 SAOs. The generation of 2 consecutive LAOs is related to the return mechanism. Indeed, after the first large excursion, which corresponds to the black or red shaded spikes in the voltage trace in the inset in the lower right panel of Fig 12, the system is projected to the left of $\xi_0$, which bounds the FN funnel, as shown for the blue and the magenta segments. This fact implies that the trajectory evolves onto $\mathcal{S}^-_{a,\epsilon}$ until it meets $\mathcal{L}^-$ at a regular jump point, from where it jumps to $\mathcal{S}^+_{a,\epsilon}$ without making SAOs. After the second LAO, the trajectory is projected into the FN funnel, following the red or the black segments, either and alternatingly between $\xi_2$ and $\xi_3$ ($R_3$), or between $\xi_3$ and $\xi_4$ ($R_4$), where 3, respectively 4, SAOs occur.

## Discussion

In this study, we developed a model of the electrical behavior of a neuronal cell subjected to different treatments interacting with the cAMP-dependent pathway linking A2aR signalling to HCN and M channel activation. The model parameters were chosen in compliance with biological constraints to ensure that the model aligns with physiological knowledge. The model qualitatively reproduced the experimental results [4, 5], validating its utility in elucidating the role of HCN and M channels. However, as the model in [4] that we build upon, the AP firing frequencies are higher than in the experimental recordings [4, 5], but, noteworthy, the relationship between the FF curves (Fig 6) agrees with the experimental results.

We found that both HCN and M channels shape the neuronal electrical activity. With enhanced HCN activation, the neuron's excitability increases, thereby facilitating the generation of APs. When the M current predominates, the resting membrane potential becomes more stable. The activation of these two currents can give rise to complex electrical phenomena such as MMOs. Our analyses strongly support the idea that both HCN and M channels are needed for producing MMOs corresponding to experimental results. Even if these temporal patterns are fragile, they are crucial for controlling the neuronal firing frequency. We thus suggest that both these two channels are involved in the signalling pathway that allows glial cells to finely tune the electrical behavior in individual neurons via activation of neuronal A2aR by ATP released from the glial cells, leading to increased cAMP in the neurons [4], which in turn activates HCN and M channels by the pathway studied in the present work, see Fig 1. Of physiological relevance, this pathway has been suggested to be involved, e.g., in mediating changes

between wake and sleep states [4]. Moreover, the reduction in firing frequency following activation of HCN or M channels has been suggested to impair working memory, in particular in response to stress [5]. A better understanding of the mechanisms underlying MMOs in neurons is thus of biomedical importance.

To explain the model behavior, we exploited the fact that we were dealing with a multiple timescale system. The 3D slow-fast subsystem of the 5D model, obtained by fixing HCN and M channel gating variables to their average values, was shown to be able to produce MMOs due to the presence of a folded node. Hence, it is not the super-slow dynamics of HCN and M

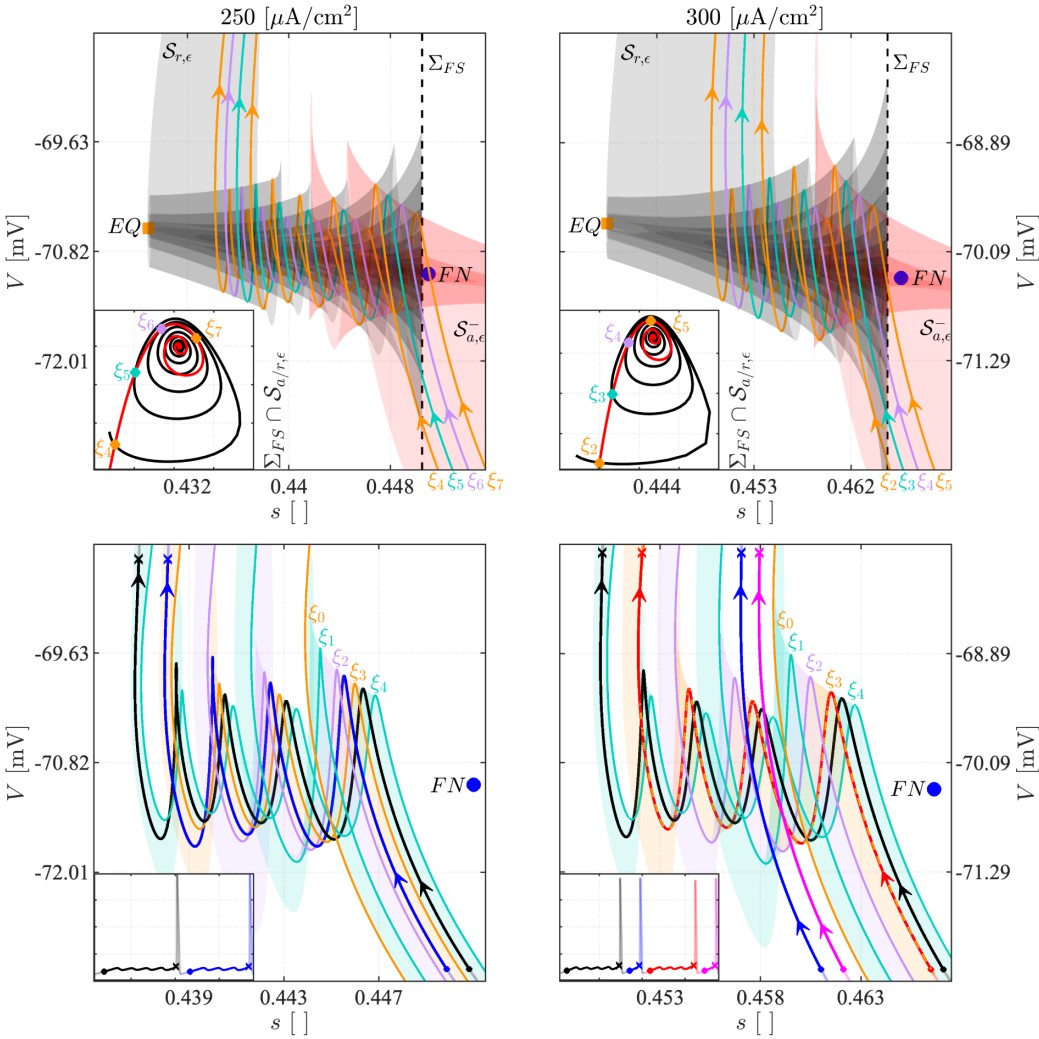

**Fig 12. Canards and attracting and repelling manifolds for the 3D Model.** The upper panels show the local reconstruction of $\mathcal{S}_{a,\epsilon}^-$ (shaded red) and $\mathcal{S}_{r,\epsilon}$ (shaded black) for $I_{App} = 250\ \mu A/cm^2$ (left) and $I_{App} = 300\ \mu A/cm^2$ (right). In each panel, the big blue dot indicates the FN. The vertical black dashed line identifies a plane $\Sigma_{FS}$ near the FN, and the inset presents the corresponding intersections between $\Sigma_{FS}$ and, respectively, $\mathcal{S}_{a,\epsilon}^-$ (red) and $\mathcal{S}_{r,\epsilon}$ (black). The colored lines $\xi_4 - \xi_7$ and $\xi_2 - \xi_5$, obtained for $I_{App}$ equal to 250 and 300 $\mu A/cm^2$, respectively, represent the canard orbits identified via the intersection points between $\mathcal{S}_{a,\epsilon}^-$ and $\mathcal{S}_{r,\epsilon}$ in the plane $\Sigma_{FS}$. In the lower panels, the canard orbits $\xi_0 - \xi_4$, the associated rotational sectors $R_1 - R_4$ and the local dynamics of the 3D model are presented for the considered values of $I_{App}$. The voltage traces (insets) and the corresponding phase-plane trajectories are decomposed into color-coded segments beginning in small circles and ending in crosses to explain how the dynamics and signatures of the MMOs are related to the identified geometrical structures.

channel activity that cause MMOs in the model, but rather, cAMP increases the average activity of these channels, setting the 3D slow-fast subsystem in a region where MMOs occur. We showed how the bifurcation diagrams for the 3D slow-fast subsystem with the modified gating variables, $\tilde{w}$ and $\tilde{r}$, provided insight into various experiments that were reproduced with simulations of the full 5D model.

Our analyses can be used to make specific predictions. For example, we predict from the 2-parameter bifurcation diagrams in Fig 8 that a moderate increase in applied current (e.g., from 250 to 300 $\mu$A/cm$^2$, i.e., from panel D to panel B in Fig 8) in a neuron producing MMOs, would lead to a transient phase with simple AP firing before MMOs reappear, since the increase in $I_{App}$ effectively shifts the region where MMOs are observed rightward (compare panel D to B in Fig 8). As a consequence, the super-slow variables $\tilde{r}$ and $\tilde{w}$ that were in the MMO region before the increase in $I_{App}$ (see panel D) fall in the LAO region just after the step in current and need to move from the LAO to the MMO region (in panel B) before MMOs reappear. On the contrary, a moderate reduction in $I_{App}$ can lead to a transient silent phase before MMOs reappear as the system moves from the white (silent) region to the MMO region, see panel D.

The 3D model qualitatively reproduced the behavior of the 5D system as presented in Fig 7, despite some discrepancies in the generated MMO signatures. We speculate that these differences are related to the assumption of constant HCN and M channel gating variables, $r$ and $w$. Reintroducing the dynamics of these variables slowly modulates the location and the properties of the folded node and its funnel, which would modify the MMO signature. For a mathematical study of this idea, numerical continuation methods may be valuable tools to continue the canard orbits of the 3D reduced model by varying the parameters $\tilde{w}$ and $\tilde{r}$. Such continuation would provide insights regarding how the FN funnel changes as the model parameters are changed. For this scope, it may be convenient to use a 4D model, such as the one obtained by constraining one of the HCN and M channel activation variables to a straight line, whilst the other is free to evolve with its dynamic. This framework could apply geometrical theories and bifurcation analyses similar to the ones used here to understand the local and global evolution of the model.

From a biological point of view, it would be interesting to investigate theoretically what happens if cAMP levels depend on the dynamics of glial cells, and how the different types of single-cell electrical phenomena impact the overall behavior of small and large networks.

In conclusion, this study employed a white-box realistic neuronal cell model to unveil the role of HCN and M channels in shaping electrical behavior as a result of cAMP signalling. We demonstrated and analyzed how the resulting currents can influence and evoke phenomena such as MMOs. This work represents a starting point for future research on the interplay between these two channels and their regulation of neuronal electrical activity, as well as for investigations on how cAMP signalling, for example resulting from interactions between neurons and glial cells, affects neuronal activity.

## Materials and methods

### Model

We build on the model presented by Lezmy et al. [4], who adapted the model by Richardson et al. [7] to pyramidal cells. The model equations are

$$
\begin{aligned}
C_m \frac{dV}{dt} &= -I_{NaF} - I_{NaP} - I_{KS} - I_L - I_{HCN} - I_M + I_{App}, \\
\frac{dp}{dt} &= \frac{p_\infty(V) - p}{\tau_p(V)}
\end{aligned}
\tag{3}
$$

where $V$ refers to the neuron's membrane potential, $t$ is the time variable, $C_m$ is the membrane capacitance, and $p \in \{h, s, r, w\}$ indicates a general activation/inactivation variable. The right-hand side of the first ODE is a sum of ionic currents. These quantify respectively the fast and persistent Na$^+$ currents ($I_{NaF}$ and $I_{NaP}$), the slow K$^+$ current ($I_{KS}$), the leakage current ($I_L$), the HCN current ($I_{HCN}$), the M-type potassium current ($I_M$) and finally the applied current ($I_{App}$). These currents follow

$$I_{NaF} = g_{NaF}\, m_\infty^3(V)\, h\, (V - E_{Na}), \tag{4}$$

$$I_{NaP} = g_{NaP}\, n_\infty^3(V)\, (V - E_{Na}), \tag{5}$$

$$I_{KS} = g_{KS}\, s\, (V - E_K), \tag{6}$$

$$I_L = g_L\, (V - E_L), \tag{7}$$

$$I_{HCN} = (g_{HCN} + cAMP\, \Delta g_{HCN})\, r\, (V - E_{HCN}), \tag{8}$$

$$I_M = (g_M + cAMP\, \Delta g_M)\, w\, (V - E_K), \tag{9}$$

with $m_\infty$, $n_\infty$, $s$, $r$, $w$ activation variables of fast Na$^+$, persistent Na$^+$, slow K$^+$, HCN and M channel currents, whereas $h$ is the $I_{NaF}$ inactivation variable. The terms $E_X$ with $X \in \{Na, K, L, HCN\}$ represent the Nernst potential for sodium, potassium, leakage and HCN currents, respectively, and $g_X$ with $X \in \{NaF, NaP, KS, L\}$, indicate the maximal whole-cell conductances. The terms $g_{HCN}$ and $g_M$ represent the maximal HCN and M channel conductances when intracellular cAMP levels are low, whereas $\Delta g_{HCN}$ and $\Delta g_M$ model their increase after a rise of the cAMP concentration, as indicated by the binary variable $cAMP$. Finally, the terms $p_\infty$ and $\tau_p$ refer to the activation/inactivation steady-state functions and time constants, respectively. For each $p \in \{h, s, r\}$,

$$p_\infty(V) = \frac{\alpha_p(V)}{\alpha_p(V) + \beta_p(V)}, \quad \tau_p(V) = \frac{1}{\phi_p(\alpha_p(V) + \beta_p(V))}, \tag{10}$$

where $\alpha_p$ and $\beta_p$ represent the transition rates of the activation/inactivation variable, and $\phi_p$ indicates the temperature correction factor which obeys the following law [26],

$$\phi_p = Q_p^{\frac{T - T_{ref}}{10}}, \tag{11}$$

where $Q_p$ and $T_{ref}$ are channel-dependent terms, and $T$ is the characteristic temperature of the experiment. The $\alpha_p$ and $\beta_p$ are given by

$$\alpha_m(V) = \frac{1.86\,(V + 25.4)}{1 - \exp[-(V + 25.4)/10.3]}, \quad \beta_m(V) = \frac{-0.086\,(V + 29.7)}{1 - \exp[(V + 29.7)/9.16]}, \tag{12}$$

$$\alpha_h(V) = \frac{-0.0336\,(V + 118)}{1 - \exp[(V + 118)/11]}, \quad \beta_h(V) = \frac{2.3}{1 + \exp[-(V + 35.8)/13.4]}, \tag{13}$$

$$\alpha_n(V) = \frac{0.186\,(V + 48.4)}{1 - \exp[-(V + 48.4)/10.3]}, \quad \beta_n(V) = \frac{-0.0086\,(V + 42.7)}{1 - \exp[(V + 42.7)/9.16]}, \tag{14}$$

$$\alpha_s(V) = \frac{0.00122\,(V + 19.5)}{1 - \exp[-(V + 19.5)/23.6]}, \quad \beta_s(V) = \frac{-0.000739\,(V + 87.1)}{1 - \exp[(V + 87.1)/21.8]}, \tag{15}$$

$$\alpha_r(V) = 0.007 \exp\left[-\frac{V + v(cAMP)}{19}\right], \quad \beta_r(V) = 0.007 \exp\left[\frac{V + v(cAMP)}{22}\right], \tag{16}$$

$$v(cAMP) = 95\,cAMP + 103.5(1 - cAMP). \tag{17}$$

Finally, the steady state ($w_\infty$) and the time constant ($\tau_w$) of the $w$ activation variable are [22]

$$w_\infty = \frac{1}{1 + \exp[-(V + 35)/10]}, \tag{18}$$

$$\tau_w = \frac{400}{3.3 \exp[(V + 35)/20] + \exp[-(V + 35)/20]}. \tag{19}$$

Table 1 presents the model parameters, obtained from [4, 7, 22]. To simulate the different experimental scenarios, the parameters presented in Table 2 were used. The ODE system was solved numerically with MATLAB [27] and XPPAUT [28]. Specifically, XPPAUT was used to derive the bifurcation diagrams presented in Figs 8 and 9, using a fourth-order Runge-Kutta method with a time step equal to 0.001 ms. MATLAB executed successive data post-processing, visualization and slow-manifold reconstruction algorithms. The system of ODEs was solved with the built-in MATLAB functions ode45 and ode78. Computer code is available at https://researchdata.cab.unipd.it/1194.

## Model reduction

We simplify the analysis of the system, and in particular of the MMOs, by exploiting the time scale separations in the model. Specifically, $V$ is fast, $h$ and $s$ are slow, and $r$ and $w$ change with a super-slow rate. Indeed, for the simulations in Fig 2, during the SAOs, the time-scale separations are $\tau_V/\tau_h < 0.03$ and $\tau_V/\tau_s < 0.004$ (while $\tau_V/\tau_h < 0.2$ during the full AP), and moreover,

**Table 1. Default model parameters.** $T$ is the temperature of the experiment, $T_{(ref,1)}$ is the reference temperature used to derive the activation/inactivation properties of the slow-K$^+$ and fast-Na$^+$ channels, while $T_{(ref,2)}$ is the reference temperature used to characterize the electrophysiological properties of the HCN channels [4].

| Parameter | Value | Unit of Measure | Parameter | Value | Unit of Measure |
|-----------|-------|-----------------|-----------|-------|-----------------|
| $g_{NaF}$ | 1000 | [mS/cm$^2$] | $T_{(ref,1)}$ | 20 | [°C] |
| $g_{NaP}$ | 1 | [mS/cm$^2$] | $T_{(ref,2)}$ | 35 | [°C] |
| $g_{KS}$ | 40 | [mS/cm$^2$] | $Q_m$ | 2.2 | [ ] |
| $g_L$ | 11.3 | [mS/cm$^2$] | $Q_n$ | 2.2 | [ ] |
| $g_{HCN}$ | 23 | [mS/cm$^2$] | $Q_h$ | 2.9 | [ ] |
| $g_M$ | 50 | [mS/cm$^2$] | $Q_s$ | 3.0 | [ ] |
| $\Delta g_{HCN}$ | 12 | [mS/cm$^2$] | $Q_r$ | 3.0 | [ ] |
| $\Delta g_M$ | 50 | [mS/cm$^2$] | $E_{Na}$ | 50 | [mV] |
| $C_m$ | 0.9 | [$\mu$F/cm$^2$] | $E_K$ | −84 | [mV] |
| $I_{App}$ | | [$\mu$A/cm$^2$] | $E_L$ | −83.38 | [mV] |
| $T$ | 37 | [°C] | $E_{HCN}$ | −50 | [mV] |

**Table 2. Translation of drug combinations applied in the various experiments into parameter sets used to simulate the model.**

| Drugs | $g_M$ [mS/cm$^2$] | $\Delta g_M$ [mS/cm$^2$] | $g_{HCN}$ [mS/cm$^2$] | $\Delta g_{HCN}$ [mS/cm$^2$] | $cAMP$ [ ] |
|---|---|---|---|---|---|
| Ctrl | 50 | 50 | 23 | 12 | 0 |
| CGS | 50 | 50 | 23 | 12 | 1 |
| ZD | 50 | 50 | 0 | 0 | 0 |
| ZD+CGS | 50 | 50 | 0 | 0 | 1 |
| XE | 0 | 0 | 23 | 12 | 0 |
| XE+CGS | 0 | 0 | 23 | 12 | 1 |
| FSK | 50 | 50 | 23 | 12 | 1 |
| FSK+Lin | 50 | 20 | 23 | 12 | 1 |

$\tau_s/\tau_r < 0.3$ and $\tau_s/\tau_w < 0.2$. To analyze the slow-fast $(V, h, s)$ subsystem, we treat $w$ and $r$ as model parameters and set them to their (approximated) average values,

$$\bar{w} = \frac{1}{\Delta T}\int_{t_0}^{t_0+\Delta T} w(t)dt \ , \quad \bar{r} = \frac{1}{\Delta T}\int_{t_0}^{t_0+\Delta T} r(t)dt \ , \quad \Delta T = T - t_0. \tag{20}$$

The simulation time interval ($T$) is set to 450 ms, which is a simulation time large enough so that the model evolves repeatedly over its stable periodic orbit, while the initial transient ($t_0$) is 350 ms. We chose this value to discard the initial transitory phase, which we observed was over well before $t = 350$ ms.

In order to be able to use the same analysis of the 3D slow-fast subsystem both in the absence and presence of cAMP, we rewrite $I_{HCN}$ and $I_M$ as follows,

$$I_{HCN} = g_{HCN}\left(1 + cAMP\frac{\Delta g_{HCN}}{g_{HCN}}\right)r\left(V - E_{HCN}\right) = g_{HCN}\,\tilde{r}\left(V - E_{HCN}\right), \tag{21}$$

$$I_M = g_M\left(1 + cAMP\frac{\Delta g_M}{g_M}\right)w\left(V - E_K\right) = g_M\,\tilde{w}\left(V - E_M\right), \tag{22}$$

where we have introduced the scaled gating variables

$$\tilde{r} = \left(1 + cAMP\frac{\Delta g_{HCN}}{g_{HCN}}\right)r \ , \quad \tilde{w} = \left(1 + cAMP\frac{\Delta g_M}{g_M}\right)w. \tag{23}$$

These super-slow scaled variables are then used as parameters in the slow-fast 3D model, which is then—for fixed $(\tilde{r}, \tilde{w})$—independent of $cAMP$. The same scaling operations are applied to map the 5D trajectories onto the $(\tilde{w}, \tilde{r})$ plane for comparison with 3D slow-fast subsystem BDs, in order to interpret the 5D model dynamics with the analyses made for the 3D model.

In conclusion, the reduced 3D model is

$$C_m\frac{dV}{dt} = -I_{NaF} - I_{NaP} - I_{KS} - I_L - I_{HCN} - I_M + I_{App}$$

$$\frac{dp}{dt} = \frac{p_\infty(V) - p}{\tau_p(V)} \ , \quad p \in \{h, s\}. \tag{24}$$

This system has itself multiple time scales, specifically, $V$ is fast and $h$ and $s$ are slow. In this formulation, $\tilde{r}$ and $\tilde{w}$ are parameters that incorporate the value of $cAMP$.

## Multiple time scale dynamical system

As noted above, the 3D model is itself a slow-fast system with two slow and one fast variable. It has the standard structure [8, 25]

$$\epsilon \frac{dx}{d\tau} = f(x, y, z), \quad \frac{dx}{dt} = f(x, y, z), \tag{25a), (25a')}$$

$$\frac{dy}{d\tau} = g(x, y, z), \quad \frac{dy}{dt} = \epsilon g(x, y, z), \tag{25b), (25b')}$$

$$\frac{dz}{d\tau} = h(x, y, z), \quad \frac{dz}{dt} = \epsilon h(x, y, z). \tag{25c), (25c')}$$

where $0 < \epsilon \ll 1$ so $x$ is the fast, and $y$ and $z$ are the slow. $t$ and $\tau = \epsilon t$ are the fast and slow time scales, respectively. The associated solutions of the ODEs systems are known as slow and fast flows, respectively. Taking the limit $\epsilon \to 0$ yields

$$0 = f(x, y, z), \quad \frac{dx}{dt} = f(x, y, z), \tag{26a), (26a')}$$

$$\frac{dy}{d\tau} = g(x, y, z), \quad \frac{dy}{dt} = 0, \tag{26b), (26b')}$$

$$\frac{dz}{d\tau} = h(x, y, z), \quad \frac{dz}{dt} = 0. \tag{26c), (26c')}$$

These are called reduced (26a–26c) and layer (26a'–26c') problems, respectively. In the former, the system evolves according to the slow flow subject to the algebraic constraint $0 = f(x, y, z)$, while in the latter, the trajectory is governed by the fast dynamics.

The critical manifold $\mathcal{C}_0$ is the set of all points $(x, y, z)$ in the phase space where $f(x, y, z) = 0$. This set represents the equilibrium points of the layer problem. The points in $\mathcal{C}_0$ are attracting (respective, repelling) if $\partial_x f$ is negative (respective, positive). Points with $\partial_x f = 0$ are said to be non-hyperbolic. The manifold $\mathcal{C}_0$ is split into attracting and repelling submanifolds, consisting of the attracting/repelling points, denoted $\mathcal{S}_a$ and $\mathcal{S}_r$, respectively. Generally, a set of non-hyperbolic points known as fold curves $\mathcal{L}$ delimits these submanifolds,

$$\mathcal{L} = \left\{ (x, y, z) \in \mathcal{C}_0 \mid \partial_x f = 0 \right\}. \tag{27}$$

For positive but small $\epsilon$, away from the fold, $\mathcal{C}_0$ perturbs to a slow manifold $\mathcal{C}_\epsilon$, which is $\mathcal{O}(\epsilon)$ close to $\mathcal{C}_0$ [25, 29]. Similarly, $\mathcal{S}_a$ and $\mathcal{S}_r$ perturb to $\mathcal{S}_{a,\epsilon}$ and $\mathcal{S}_{r,\epsilon}$. This relaxation perturbs the solutions of the reduced and the layer problems. Specifically, the trajectory is dominated by the slow (respective, fast) dynamics when it is close to (respective, away from) $\mathcal{C}_0$. A method to understand the dynamics near the fold, where the system becomes singular, consists in deriving the desingularized model. This model results from applying the total differentiation theorem, using the explicit expression of $\mathcal{C}_0$, and performing a time-rescaling known as desingularization, which yields [8, 25],

$$\begin{aligned} \frac{dx}{dt} &= \partial_y f\, g + \partial_z f\, h, \\ \frac{dz}{dt} &= -\partial_x f\, h. \end{aligned} \tag{28}$$

where $\mathcal{C}_0$ is expressed (locally) as $y = \gamma(x, z)$. Singularities of the desingularized system that satisfy $g(x, \gamma(x, z), z) = h(x, \gamma(x, z), z) = 0$ are also equilibria of the non-desingularized system and are called ordinary singularities. Instead, folded singularities satisfy $\partial_x f = 0$ and $\partial_y fg + \partial_z fh = 0$. The first condition imposes that the point lies on $\mathcal{L}$. A folded singularity is classified as a folded node (FN), folded saddle, or folded foci according to the eigenvalues of the Jacobian of the desingularized system. The solution of the desingularized system passing tangentially to the eigendirection of the FN associated with the eigenvalue with the highest modulus is known as the *strong canard*. When the singular limit $\epsilon \to 0$ is relaxed, the strong canard persists and bounds, together with the fold line $\mathcal{L}$, a region on $\mathcal{S}_{a,\epsilon}^-$, known as the *funnel*, where trajectories experience SAOs. These can lead to MMOs when combined with an appropriate return mechanism [8]. Some special solutions, known as *secondary canards*, cross $\mathcal{L}$ with non-zero speed and evolve close to $\mathcal{S}_{r,\epsilon}$ for a finite amount of time. These secondary canards organize the FN funnel into rotational sectors $R_i$, where $i$ denotes the number of SAOs of orbits starting in $R_i$. We note that any solution of the system of ODEs that crosses $\mathcal{L}$ with non-zero speed and follows the repelling slow manifold $\mathcal{S}_{r,\epsilon}$ for a finite amount of time is a *canard*. However, in this paper, when we use the term *canard* we will refer to the strong and the secondary canards only.

For the model analyzed here, the critical manifold can be written

$$\mathcal{C}_0 = \left\{ (V, h, s) \in \mathbb{R}^3 \,\middle|\, s = \gamma(V, h) \right\}, \qquad \gamma(V, h) = \gamma_0 + \gamma_r^t + \gamma_w^t + \gamma_r^\theta, \tag{29}$$

where the last expression presents a decomposition to understand better how variation in $(\tilde{w}, \tilde{r})$ moves $\mathcal{C}_0$. The terms are

$$\gamma_0 = -\frac{I_{NaF}(V, h) + I_{NaP}(V) + I_L(V) - I_{App}}{g_{KS}(V - E_K)}, \tag{30}$$

$$\gamma_r^t = -\frac{g_{HCN}}{g_{KS}}\tilde{r}, \qquad \gamma_w^t = -\frac{g_M}{g_{KS}}\tilde{w}, \qquad \gamma_r^\theta = \frac{g_{HCN}}{g_{KS}}\frac{E_{HCN} - E_K}{V - E_K}\tilde{r}. \tag{31}$$

Here $\gamma_0$ defines the basic structure of $\mathcal{C}_0$ when both channels are inhibited. Instead, $\gamma_r^t$ and $\gamma_w^t$ are shifting factors, representing how activation of HCN and M channels move $\mathcal{C}_0$. Finally, $\gamma_r^\theta$ applies a shift inversely proportional to $V$, but still proportional to $\tilde{r}$.

To split $\mathcal{C}_0$ into its submanifolds, the fold $\mathcal{L}$ is computed,

$$\mathcal{L} = \left\{ (V, h, s) \in \mathcal{C}_0 \,\middle|\, h = \psi(V) \right\}, \qquad \psi(V) = \psi_0 + \psi_r. \tag{32}$$

Again, the definition decomposes the fold line to evaluate how modification of HCN and M channel activation move $\mathcal{L}$,

$$\psi_0 = -\frac{[(V - E_K)\partial_V I_{NaP} - I_{NaP}] + g_L(E_L - E_K) + I_{App}}{(V - E_K)\partial_V(g_{NaF}\, m_\infty^3(V)\,(V - E_{Na})) - g_{NaF}\, m_\infty^3(V)\,(V - E_{Na})}, \tag{33}$$

$$\psi_r = -\frac{g_{HCN}(E_{HCN} - E_K)}{(V - E_K)\partial_V(g_{NaF}\, m_\infty^3(V)\,(V - E_{Na})) - g_{NaF}\, m_\infty^3(V)\,(V - E_{Na})}\tilde{r}. \tag{34}$$

We see that the HCN gating variable $\tilde{r}$ translates the fold line, whereas the $h$ component of $\mathcal{L}$ is M channel ($\tilde{w}$) independent. However, because the fold is constrained to live on $\mathcal{C}_0$ by construction, the $s$ component of $\mathcal{L}$ does depend on $\tilde{w}$ (as well as on $\tilde{r}$) through $\gamma$, which considers the shift caused by the M channel activity.

Finally, the desingularized 3D model is

$$
\frac{dV}{dt} = \partial_h I_{NaF} \frac{h - h_\infty(V)}{\tau_h(V)} + \partial_s I_{KS} \frac{\gamma(V, h) - s_\infty(V)}{\tau_s(V)},
$$

$$
\frac{dh}{dt} = -\frac{h - h_\infty(V)}{\tau_h(V)} \sum_{i \in X} \partial_V I_i, \tag{35}
$$

$$
X = \{NaF, NaP, KS, L, M, HCN\}.
$$

This system has two types of singularities. The ordinary singularities are obtained as the numerical solutions of $\gamma(V, h_\infty(V)) = s_\infty(V)$. These points correspond to the equilibriums of the non-desingularized 3D model. Instead, the folded singularities are obtained by considering those points belonging to $\mathcal{L}$ where

$$
\partial_h I_{NaF}[\psi(V) - h_\infty(V)]\tau_s(V) = \partial_s I_{KS}[s_\infty - \gamma(V, \psi(V))]\tau_h(V). \tag{36}
$$

Solving this equation numerically provides the $V$ coordinates of those points in $\mathcal{L}$ classifiable as folded singularities. Numerically, we found that the desingularized system possesses a folded node singularity.

To reconstruct the slow manifold shown in Fig 12, we exploit an idea similar to the one presented in [30], see also [15]. It relies on evolving in forward and backward time the system starting on a grid of initial conditions along two lines located in $\mathcal{S}_a^-$ and $\mathcal{S}_a^+$ of $\mathcal{C}_0$, respectively, to obtain a set of curves lying in, respectively, a strip of $S_{a,\epsilon}^-$ and $S_{r,\epsilon}$. The solutions of the system are terminated when they reach a plane $\Sigma_{FS}$ perpendicular to the $s$-axis and slightly perturbed from the position of the folded node. The forward and backward solutions are then projected onto this plane to find the intersections between the repelling and the attracting slow manifolds, which correspond to canard orbits. The remaining canards were computed with brute force, exploiting the fact that the canards of interest are those trajectories separating the FN funnel in different rotational sectors.

## Author Contributions

**Conceptualization:** Morten Gram Pedersen.

**Formal analysis:** Matteo Martin, Morten Gram Pedersen.

**Investigation:** Matteo Martin, Morten Gram Pedersen.

**Methodology:** Matteo Martin, Morten Gram Pedersen.

**Project administration:** Morten Gram Pedersen.

**Software:** Matteo Martin.

**Supervision:** Morten Gram Pedersen.

**Visualization:** Matteo Martin.

**Writing – original draft:** Matteo Martin, Morten Gram Pedersen.

**Writing – review & editing:** Matteo Martin, Morten Gram Pedersen.

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
