## [Decision Letter · Decision Letter 0]

15 Nov 2023

Dear Dr. Pedersen,

Thank you very much for submitting your manuscript "Modelling and analysis of cAMP-induced mixed-mode oscillations in cortical neurons: Critical roles of HCN and M-type potassium channels" for consideration at PLOS Computational Biology.

As with all papers reviewed by the journal, your manuscript was reviewed by members of the editorial board and by several independent reviewers. In light of the reviews (below this email), we would like to invite the resubmission of a significantly-revised version that takes into account the reviewers' comments.

Reviewer 2 correctly suggests that the abbreviation "RMP" for a neuron's resting potential is not widely known in the likely readership of PLoS CB. Nevertheless, instead of "V_m", which normally refers to the (dynamic) value of the membrane potential, a more clear abbreviation would be "V_rest". 

We cannot make any decision about publication until we have seen the revised manuscript and your response to the reviewers' comments. Your revised manuscript is also likely to be sent to reviewers for further evaluation.

Sincerely,

Boris S. Gutkin

Academic Editor

PLOS Computational Biology

Lyle Graham

Section Editor

PLOS Computational Biology

Reviewer's Responses to Questions

**Comments to the Authors:**

Reviewer #1: Review attached.

Reviewer #2: I was happy to see this paper appear as a submission to PLoS Computational Biology. I think it provides an excellent example of using ideas from the theory of multiple timescale dynamical systems to explain experimental observations in neural models that include the contributions of specific ion channels, and it is important to have this type of work disseminated to a broad audience such as that of PLoS CB. I do have a variety of comments aimed at improving how well the paper conveys its key points, at making it more suitable for the PLoS CB readership, and at making a few minor corrections. I will present these in order of appearance in the paper:

RESULTS

1) Start of 1st section: Some clarification is needed: What types of neurons were used in the experiments in refs. [4], [5]? What types of neurons is the model in this paper meant to describe? Do these match?

2) Top of par 2 of 2nd section: Need some sort of reference ahead to the Methods so that readers know what is meant by "The system of ODEs". Also, please give a brief summary of the model components here, for readers who don't want to look ahead to Methods at this point.

3) RMP is not a standard abbreviation for resting membrane potential. Better to use V_m.

4) Line 79: the phrase "...increase the comprehension of..." is awkward; rephrase.

5) *** The organization of the simulation results and comparisons to experiments is more confusing than it needs to be, in my opinion. I think the authors should re-organize so that results are presented as follows. First, the baseline dynamics without any manipulation should be shown, across the relevant range of Iapp values. Second, the effects of M block and, separately, HCN block, when applied to the baseline conditions, should be shown. Third, the authors should show what happens when cAMP is increased or activated, again over a range of Iapp. Finally, the authors should show what happens when M block and HCN block are applied after cAMP is increased. In all of these instances, the authors should make clear where (in parameter space) there are regular APs and where there are MMOs. This is related to Figure 6 but should come earlier and be presented in this more organized, systematic way.

6) Line 106: I believe "leftward" should be replaced with "rightward".

7) *** The authors introduce confusion by referring to (V,h,s) as the "fast subsystem" while also noting that h, s are slower than V and stating that (V,h,s) is itself a "slow-fast system". Perhaps they were skittish about explicitly referring to 3 timescales, but I think that would be the better way to go. So w, r will be superslow; h, s will be slow; and V will be fast. It will be necessary to come up with a name for the (V,h,s) system. I think that calling that the slow-fast subsystem would be fine. Or the authors could call it the layer system.

8) The equation where tilde(r) is introduced has a typo (g_HNC -> g_HCN). In the same line of the paper, more explanation is needed about how to interpret the Delta g terms.

9) *** Related to the previous comment: I find the use of tilde(r) and tilde(w) confusing. Why can't the authors just vary w and r? If they are going to use tilde(w) and tilde(r), they should also state explicitly how they show up in the model equations. Currently, for example, the authors state above line 346 that they introduce scaling factors and define tilde(w) and tilde(r), but once these terms are defined, the authors do not state how the V equation is adjusted to include these terms.

10) Lines 197-8: Provide more details about which aspects of Fig. 6 are explained here; "This explains the results shown in Fig 6..." is not adequate. For example, Fig. 6 shows results about both H and HCN activation via increased cAMP, which is not discussed in the earlier part of this paragraph.

11) Fig. 9: If Delta gM=0, then the elevated cAMP case is actual identical to the control case, so why don't the blue and magenta curves meet at Delta gM=0 (similary for red and black)? Is this because of HCN? Please clarify. Also, please mention in the caption what value of cAMP is used here.

12) Line 231: same issue with fast vs. slow-fast as mentioned above.

13) Please add a new 3-d figure showing the surface C0 along with the folds L and the various S sheets.

14) Line 263: You cannot assume that PLOS CB readers are familiar with canards.

15) Please provide more information about Fig. 10 to make it clear why the CGS case with Iapp=250 gives 1 LAO per cycle, while Iapp=300 gives >1 LAOs.

16) Inset of Fig. 11, upper right: Why are there sharp corners on the trajectory?

17) Fig. 11: Please tell us how the xi_i are computed. Also, more explanation and labeling is needed so that readers can see that the trajectory is projected to the left of xi_0 after the 1st return for Iapp=300.

18) *** Discussion: The authors should say more about the possible functional role of the MMOs that they discuss. They should also state some explicit predictions from their multiple timescale analysis.

19) Methods: How can g_HCN and g_M be called "maximal" conductances when the Delta terms are positive?

20) Table 1: How can you have two different temperatures for 2 different current gates in the same model of the same cell? Also, why is E_L specified to the hundreths place while none of the other parameters are? Is the model extremely sensitive to E_L? Finally, why is E_L so negative? This is an unusually low value, in my experience.

21) Model reduction: The authors should provide a quantitative justification for the statement that h, s are slower than V.

22) Pg. 17, top: The authors should provide some justification for the values that they choose for T, t0, T-t0.

I look forward to seeing the revisions! I certainly hope that the authors can make these changes and improvements. I would love to use a published version of this work in my graduate mathematical neuroscience course.

**Have the authors made all data and (if applicable) computational code underlying the findings in their manuscript fully available?**

Reviewer #1: Yes

Reviewer #2: **No: **The authors state that they used XPPAUT and MATLAB for their computations. They should share code for the neural model itself as well as for finding and integrating the relevant canard solutions.

PLOS authors have the option to publish the peer review history of their article (what does this mean?). If published, this will include your full peer review and any attached files.

Reviewer #1: **Yes: **Lou Zonca

Reviewer #2: No
---

## [Decision Letter · Decision Letter 1]

10 Mar 2024

Dear Dr. Pedersen,

We are pleased to inform you that your manuscript 'Modelling and analysis of cAMP-induced mixed-mode oscillations in cortical neurons: Critical roles of HCN and M-type potassium channels' has been provisionally accepted for publication in PLOS Computational Biology.

Best regards,

Boris S. Gutkin

Academic Editor

PLOS Computational Biology

Lyle Graham

Section Editor

PLOS Computational Biology

Reviewer's Responses to Questions

**Comments to the Authors:**

Reviewer #1: It's fine for me, thanks for the work!

Just one thing: the link you give for the codes download at the beginning of your reply does not work. However, the one in the Methods section is correct..

Reviewer #2: I apologize for keeping the authors waiting for such a brief review, but: I think the authors have done a good job revising the manuscript to address my comments and explaining to me, in their responses, the reasoning behind their revisions. I think the new figures and figure clarifications are especially helpful.

**Have the authors made all data and (if applicable) computational code underlying the findings in their manuscript fully available?**

Reviewer #1: Yes

Reviewer #2: Yes

PLOS authors have the option to publish the peer review history of their article (what does this mean?). If published, this will include your full peer review and any attached files.

Reviewer #1: **Yes: **Lou Zonca

Reviewer #2: No

---

## [Editor Report · Acceptance letter]

19 Mar 2024

PCOMPBIOL-D-23-01567R1 

Modelling and analysis of cAMP-induced mixed-mode oscillations in cortical neurons: Critical roles of HCN and M-type potassium channels

Dear Dr Pedersen,

I am pleased to inform you that your manuscript has been formally accepted for publication in PLOS Computational Biology. Your manuscript is now with our production department and you will be notified of the publication date in due course.

With kind regards,

Zsofia Freund
